**Data Availability Statement:** EPHIA 2017-2018 public release data is posted on the ICAP Population-Based HIV Impact Assessment data

# Progress towards controlling the HIV epidemic in urban Ethiopia: Findings from the 2017–2018 Ethiopia population-based HIV impact assessment survey

Sileshi Lulseged[1]◉*, Zenebe Melaku[1]‡, Abebe Habteselassie[2]‡, Christine A. West[3]◉, Terefe Gelibo[1]◉, Wudinesh Belete[2]‡, Fana Tefera[4]‡, Mansoor Farahani[5]◉, Minilik Demissie[2]‡, Wondimu Teferi[4]‡, Saro Abdella[2]‡, Sehin Birhanu[3]‡, Christine E. Ross[4]◉, the EPHIA Study Group¶

1 ICAP in Ethiopia, Mailman School of Public Health, Columbia University, Addis Ababa, Ethiopia, 2 Department of HIV and TB, Ethiopia Public Health Institute, Addis Ababa, Ethiopia, 3 Division of Global HIV and TB, Centre for Global Health, Centres for Disease Control and Prevention, Atlanta, Georgia, United States of America, 4 Care and Treatment Branch, United States Centres for Disease Control and Prevention, Addis Ababa, Ethiopia, 5 ICAP, Mailman School of Public Health, Columbia University, New York, New York, United States of America

◉ These authors contributed equally to this work.
‡ ZM, AH, WB, FT, MD, WT, SA and SB also contributed equally to this work.
¶ List of the EPHIA Study Group is provided in the Acknowledgements and also available from: https://phia.icap.columbia.edu/wpcontent/uploads/2020/11/EPHIA_Report_280820_High-Res.pdf.
* sileshilulseged@gmail.com

## Abstract

### Introduction

In 2014, the Joint United Nations Programme on HIV/AIDS set an 'ambitious' 90-90-90 target for 2020. By 2016, there were disparities observed among countries in their progress towards the targets and some believed the targets were not achievable. In this report, we present the results of data from the Ethiopia Population-based HIV Impact Assessment survey analyzed to assess progress with the targets and associated factors.

### Methods

We conducted a nationally representative survey in urban areas of Ethiopia. Socio-demographic and behavioural data were collected from consenting participants using a structured interview. HIV testing was done following the national HIV rapid testing algorithm and seropositivity confirmed using a supplemental laboratory assay. HIV viral suppression was considered if the viral load was <1,000 RNA copies/ml. Screening antiretroviral drugs was done for efavirenz, lopinavir, and tenofovir, which were in use during the survey period. In this analysis, we generated weighted descriptive statistics and used bivariate and logistic regression analysis to examine for associations. The 95% confidence interval was used to measure the precision of estimates and the significance level set at p<0.05.

website (https://phia-data.icap.columbia.edu/datasets?country_id=12). Dataset documentation is available for immediate download and datasets are available upon request by registering for an account and submitting the request form.

**Funding:** NO. The funders had no role in study design, data collection and analysis, decision to publish, or preparation of the manuscript.

**Competing interests:** I have read the journal's policy and the authors of this manuscript have the following competing interests. None of the authors have any conflict on interest.

## Results

Of 19,136 eligible participants aged 15–64 years, 614 (3% [95% CI: 0.8–3.3]) were HIV-positive, of which 79.0% (95% CI: 4.7–82.7) were aware of their HIV status, and 97.1% (95% CI: 95.0–98.3 were on antiretroviral therapy, of which 87.6% (95% CI: 83.9–90.5) achieved viral load suppression. Awareness about HIV-positive status was significantly higher among females (aOR = 2.8 [95% CI: 1.38–5.51]), significantly increased with age, the odds being highest for those aged 55–64 years (aOR = 11.4 [95% CI: 2.52–51.79]) compared to those 15–24 years, and was significantly higher among those who used condom at last sex in the past 12 months (aOR = 5.1 [95% CI: 1.68–15.25]). Individuals with secondary education and above were more likely to have achieved viral suppression (aOR = 8.2 [95% CI: 1.82–37.07]) compared with those with no education.

## Conclusion

Ethiopia made encouraging progress towards the UNAIDS 90-90-90 targets. The country needs to intensify its efforts to achieve the targets. A particular focus is required to fill the gaps in knowledge of HIV-positive status to increase case identification among population groups such as males, the youth, and those with low education.

## Introduction

In 2014, the Joint United Nations Programme on HIV/AIDS launched the ambitious 90-90-90 treatment targets to end the HIV pandemic by 2020; the aim was to diagnose 90% of all HIV-positive persons, provide antiretroviral treatment (ART) to 90% of those diagnosed, and achieve viral suppression in 90% of those treated [1]. When these treatment targets were achieved, at least 73% (90*90*90) of all people living with HIV (PLHIV) worldwide would achieve viral load suppression (VLS). Modelling suggested that achieving these targets by 2020 would enable the world to continue with actions that will end the HIV/AIDS epidemic by 2030, which in turn would generate profound health and economic benefits [1, 2]. The targets were widely promoted and adopted by countries and international implementing partners [3]. By 2020, though only a few countries had taken sufficient action to achieve the targets, much of the complex reality of the HIV epidemic was increasingly understood and countries were in a better position to identify gaps and develop strategies to reach PLHIV who were left behind [4]. A comprehensive approach would be required to fill the gaps and achieve the next set of UNAIDS 95-95-95 testing and treatment targets; 95% of people living with HIV know their HIV status, 95% of PLHIV who know their status are on treatment, and 95% of people on treatment have suppressed viral loads, thereby reducing the annual number of new HIV infections among adults to 200,000 or fewer; and achieving zero discrimination by 2030 [5].

Ethiopia introduced a large-scale ART programme in 2005 and made remarkable progress by 2016 in reducing the prevalence of HIV infection from 1.4% to 0.9% and the number of AIDS-related deaths from 83,000 to 15,600 [6]. The country achieved a 50% reduction in HIV incidence rate and a decline in HIV mortality rate, reaching a tipping point (the incidence mortality ratio, where the number of new infections drops below the number of deaths) [7]. However, after the remarkable decline in the incidence and prevalence of HIV infection in all age groups, progress started to slow down in the last few years. Ethiopia was not able to fully achieve the 90-90-90 targets due to several barriers [8]. Economic constraints, perceived

stigma and discrimination, fasting, holy water, medication side effects, and dissatisfaction with healthcare services were major reasons for non-adherence and interruption of treatment among PLHIV [9]. Isolated studies also showed that treatment failure is an emerging barrier that is associated with transmission of drug-resistant HIV strains at the population level [10].

Reports on HIV care and treatment services were primarily drawn from the Ethiopian demographic and health survey (EDHS),supplemented by data from routine antenatal sentinel surveillance, programme monitoring, mathematical modeling, and special studies in tracking the magnitude and dynamics of the HIV epidemic. There were no national-level, population-based studies conducted in Ethiopia, which measured ART coverage and VLS. Reliable nationally representative data for HIV programming was lacking. Therefore, as part of a multi-country population-based HIV impact assessment (PHIA) project, the government of Ethiopia spearheaded the Ethiopia population-based HIV impact assessment (EPHIA) survey to measure the Ethiopia's national response to the epidemic, assess the coverage and impact of HIV services, and measure HIV-related risk behaviors. We are reporting here the results of an analysis of EPHIA survey data, focusing on the progress made towards the UNAIDS 90–90–90 targets and associated factors to inform the national policy and HIV programming that will help the country achieve the 95-95-95 targets.

## Materials and methods

The survey protocol was approved by the institutional review boards of the Ethiopian Research Institute, Addis Ababa (EPHI-IRB-028-2017), Columbia University, City of New York, USA (IRB-AAAR5279), and US Centers for Disease Control and Prevention, Atlanta, USA (Protocol #7044).

### Survey setting and population

Ethiopia is the second-most populous country in Africa and is divided into nine regional states and two city administrations. Projections from the 2007 housing and population census estimated the total population for the year 2017 at 105 million and the breakdown by age showed that 47% of the population were under 15 years of age, 49% between 15 and 64 years, and 4% were 64 years and above [11]. The median age was 16.8 years and the population growth rate estimated at 2.6%, the eighth highest in the world [12]. In the EPHIA survey, geographic classification followed the 2007 National Population and Housing Census, which included all capitals of the nine regional states and the two city administrations, zones, and woredas (districts) where inhabitants were primarily engaged in non-agricultural activities [13]. Only 25% of Ethiopia's population resided in urban areas, but population density and consequently the density of HIV infection were high in these parts of the country [14]. HIV prevalence was very low and HIV cases were thinly spread out across the rural areas, which constitute over 80% of the landmass of the country [7]. Consequently, we conducted the EPHIA survey in urban Ethiopia.

### Study design

The EPHIA was a household (HH)-based survey conducted among urban adults 15–64 years and children 0–14 years of age. The survey employed the methods used by the multi-country population-based HIV impact assessment (PHIA) survey conducted in other 14 African countries [15]. The details of the design, methods, and tools (including the original English version of the questionnaire) are available in the survey report on the ICAP Columbia University website [16]. The sample size was powered to provide estimates of national and regional levels of VLS among HIV positive adults. All adult HH members were included in the HH rosters compiled for sampling purposes. The selection of participants involved compiling a list of all

individuals known to reside in the HH or who slept in the HH during the night prior to data collection, identifying rostered individuals who were eligible for data collection, selecting those meeting the age and residency requirements of the study.

## Data collection

The data were collected from October 2017 to April 2018. Questionnaire and field laboratory data were collected on mobile Google Nexus 9 tablets. The adult questionnaires were administered in Amharic, Oromiffa, Tigrigna, Afarigna, and Somaligna (the five languages commonly used in Ethiopia) to all eligible participants aged 15 years and older through face-to-face interviews. The data included demographic, behavioural, and clinical information and participants' self-reported knowledge of HIV and treatment status. Home-based HIV rapid testing was done using the Ethiopian national HIV rapid testing algorithm and CD4 T-cell counts determined on PIMA machine. Laboratory evaluation also included Geenius confirmatory testing, plasma ARV drugs, and genotyping for HIV transmitted drug resistance mutations done at the central level by the Ethiopian Public Health Institute (EPHI). Samples from all confirmed HIV-positive participants were evaluated for the presence of three ARVs (efavirenz, lopinavir, and tenofovir), which were selected as markers for the most prescribed first- and second-line regimens in use at the time of the survey. Samples from participants who had suppressed viral loads and/or reported being on ART, but had no evidence of the first three compounds, were tested for nevirapine.

## Data analysis

The 90-90-90 target achievements in this analysis were based on self-reports adjusted for ARV drug detection. 'Awareness' and 'being on treatment' were adjusted such that individuals with detected ARVs were classified as 'aware' (first 90) and 'on treatment' (second-90) even if they did not self-report knowing their HIV-positive status or being on ART. The achievements for the second-90 were expressed either as conditional (proportion on treatment among those who were aware) or unconditional (proportion on treatment among all positives). For the third-90, achievements were expressed as conditional (proportion with viral suppression among those on treatment) or unconditional (proportion with viral suppression among all positives). Overall, HIV-positive participants who were aware of their HIV status, receiving ART, and had HIV-1 RNA <1000 copies per mL were considered to have VLS. Overall cascade response was compared against the UNAIDS cascade response of 73% (i.e. 90*90*90).

The results of the analysis reported here are weighted estimates unless otherwise stated. The weighting accounted for sample selection probabilities adjusted for non-response and non-coverage. Non-response adjusted weights were calculated for HHs, individual interviews, and individual blood draws in a hierarchical manner. Nonresponse adjustments for the initial individual and blood-level weights were based on the development of weighting adjustment cells defined by a combination of variables that are potential predictors of response and HIV status. The weighted estimates were produced by using the Jackknife replication, a method that estimates the variance/standard and bias of a large population sample data by involving a leave-one-out strategy of the estimation of a parameter in a dataset [17] done on Stata 14 [18].

We generated weighted descriptive statistics for the variables and included in the analysis and determined the association between the explanatory and outcome variables using a bivariate analysis with Chi-square statistic to test for significance of the association between categorical variables. Multiple logistic regression was employed to examine for an independent association between explanatory variables and the target outcomes. Variables with p-value of 0.2 in the bivariate analysis were selected for the multivariate regression analysis. The 95%

confidence interval was used to measure the precision of point estimates and a p-value of <0.05 considered statistically significant.

### Ethical considerations

The survey team informed all potential participants that participation was voluntary and that they did not need to disclose personal information, which they were uncomfortable sharing, and that they could withdraw from the survey at any time. The team also provided to all potential participants a printed copy of the consent form in one of the six survey languages depending upon their preference prior to initiation of any survey procedures. For illiterate participants, an impartial witness chosen by the participant was involved. Potential participants who did not speak any of the six survey languages were considered ineligible. A designated HH head provided written consent for HH members to participate in the survey, after which individual members were rostered during a HH interview.

Consent was then obtained from participants aged 18–64 years who were willing and able to provide written consent for the interview and biomarker components of the survey, including home-based testing and counselling with the return of HIV-test results. Emancipated minors constituting participants aged 13–17 years and were working or earning their living, married, or parenting [19] provided the written consent like the age group 18–64 years. The waiver of parental/guardian consent for the emancipated minors was given as part of the protocol approval by the institution review boards. Survey teams ensured that written parental/ guardian permission was obtained for assenting minors aged 15–17 years. At each stage of the process, the participant indicated consent by signing or making a mark on the consent form on the tablet and in a printed copy, which was retained by the participant. Receipt of tests results was a requirement for participation in the biomarker component.

## Results

### Participants' characteristics

A total of 21,560 eligible adults aged 15–64 years was identified from among 25,416 adult HH members who were randomly selected proportional to the population structure from among 11,841 HH in 393 randomly selected enumeration areas (EAs) (Fig 1). Of a total of 25,416 adults >15 years and above rostered during the HH interview, 21, 560 were eligible, of which 20,170 (12,158 females and 8,012 males) were interviewed and 19,136 (11,599 females and 7,537 males) had blood specimens drawn for biomarker testing. Of 3,856 (6.4%) HH members who were not eligible, 2,516 were not in the HH during the previous night, 1,338 were excluded because of age over 64 years, cognitive impairment or intellectual disability, language and other reasons, and only two were rostered in error.

Most of the participants (92.9%) were from the Oromia, Amhara, Addis Ababa, SNNPR and Tigray regions (Table 1). Over two-thirds (67.9%) of them were female and 34.9% were in the age group 5 to 24 years. Participants who were married or living together constituted 52.1%, those who had primary education 35.5%, and those not employed in the past 12 months 52.9%. A majority (95.0%) had first sex after 15 years of age, and about two-thirds (68.2%) had one sexual partner in the last 12 months prior to the survey, 65.2% did not use a condom at last sex in the past 12 months and 70.5% reported ever being tested for HIV.

### HIV prevalence

Of the 19,136 participants who were tested for HIV, 614 tested positive, an overall HIV prevalence of 3% (95% CI: 2.8–3.3), which varied by region from 0.8% (95% CI: 0.4–1.6) in Somali

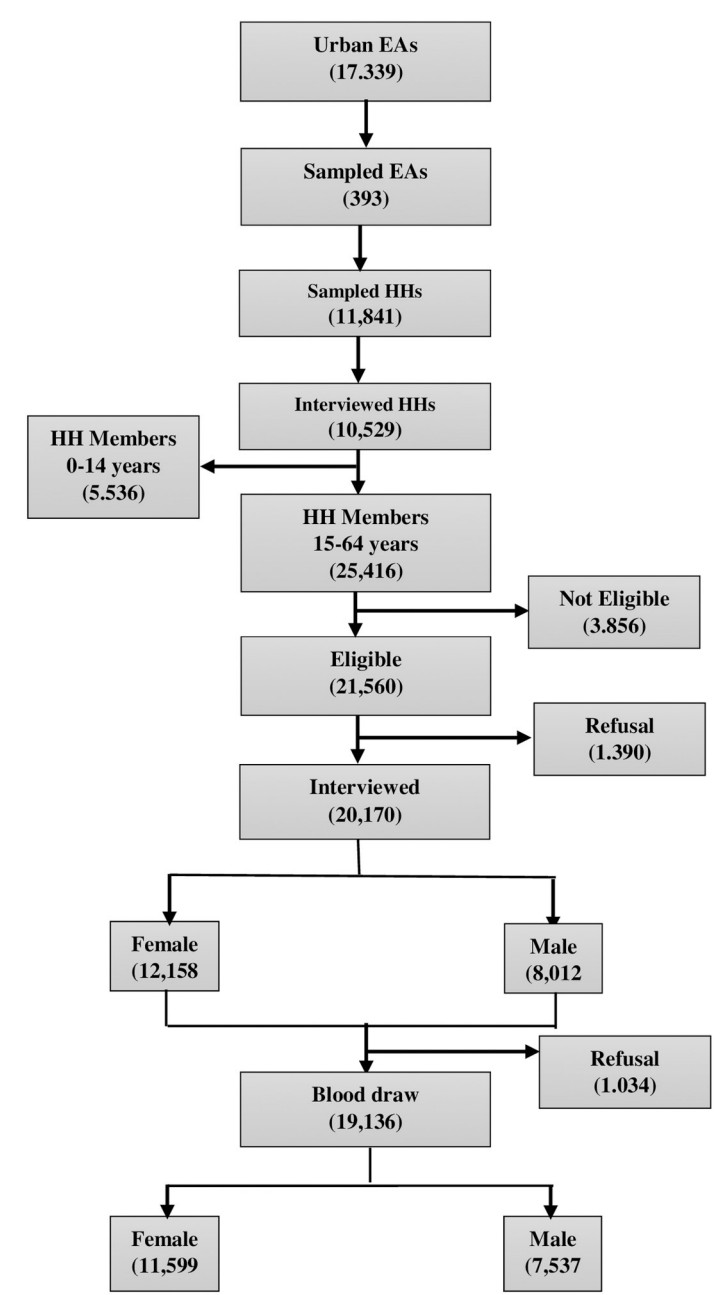

**Fig 1. Sampling flow chart, Ethiopia population-based HIV impact assessment survey 2017–2018.**

to 5.7% (95% CI: 4.2–7.7) in Gambella (p<0.0001). The prevalence was significantly higher among females than males (4.1% [95% CI: 3.7–4.5]) vs 1.9% (95% CI: 1.6–2.3) (p<0.0001) and was higher (6.2% [95% CI: 5.4–7.2]) in the age group 35–44 years, among those who were widowed (14.7% [95%CI = 12.1–17.7]), those with no formal education (5.2% [95% CI: 4.2–6.3]), female-headed HHs (4.0% [95% CI: 3.6–4.5]), and food-insecure (5.0% [95% CI: 3.7–6.8]) than in the other the categories in their respective groups. The prevalence was also higher among those who had first sexual exposure before age of 15 years (6.8% [95% CI: 5.3–8.7]),

**Table 1. HIV prevalence by socio-demographic and behavioural characteristics among participants aged 15–64 years, Ethiopia population-based HIV impact assessment, 2017–2018.**

| Characteristics | N (Weighted %) | HIV Infection Status | | | | P-value |
|---|---|---|---|---|---|---|
| | | Positive | | Negative | | |
| | | n | Weighted % (95% CI) | n | Weighted % (95% CI) | |
| **Administrative Region** | | | | | | |
| Tigray | 1,369 (7.5%) | 39 | 2.7 (1.9–3.6) | 1,330 | 97.3 (96.4–98.1) | <0.0001 |
| Afar | 821 (1.3%) | 32 | 4.1 (2.9–5.8) | 789 | 95.9 (94.2–97.1) | |
| Amhara | 2,999 (18.2%) | 118 | 4.1 (3.4–5.0) | 2,881 | 95.9 (95.0–96.6) | |
| Oromia | 4,510 (34%) | 149 | 3.0 (2.5–3.6) | 4,361 | 97.0 (96.4–97.5) | |
| Somali | 926 (1.3%) | 8 | 0.8 (0.4–1.6) | 918 | 99.2 (98.4–99.6) | |
| Benishangul Gumuz | 798 (1.3%) | 20 | 2.4 (1.6–3.8) | 778 | 97.6 (96.2–98.4) | |
| SNNPR | 2,665 (16.2%) | 49 | 1.8 (1.3–2.4) | 2,616 | 98.2 (97.6–98.7) | |
| Gambella | 788 (0.6%) | 44 | 5.7 (4.2–7.7) | 744 | 94.3(92.3–95.8) | |
| Harari | 697 (0.7%) | 32 | 4.6 (3.3–6.5) | 665 | 95.4 (93.5–96.7) | |
| Addis Ababa | 2,780 (17.7%) | 88 | 3.1 (2.5–3.8) | 2,692 | 96.9 (96.2–97.5) | |
| Dire Dawa | 783 (1.2%) | 35 | 4.6 (3.3–6.5) | 748 | 95.4 (93.5–96.7) | |
| Total | 19,136 | 614 | 3.0 (2.8–3.3) | 18,522 | 97.0 (96.7–97.2) | |
| **Sex** | | | | | | |
| Female | 11,599 (50.5%) | 461 | 4.1 (3.7–4.5) | 11,138 | 95.9 (95.5–96.3) | <0.0001 |
| Male | 7,537 (49.5%) | 153 | 1.9 (1.6–2.3) | 7384 | 98.1 (97.7–98.4) | |
| Total | 19,136 | 614 | 3.0 (2.8–3.3) | 18,522 | 97.0 (96.7–97.2) | |
| **Age group** | | | | | | |
| 15–24 years | 7,547 (34.9%) | 62 | 0.7 (0.5–1.0) | 7,485 | 99.3 (99.0–99.5) | <0.0001 |
| 25–34 years | 5,664 (30.3%) | 175 | 2.6 (2.2–3.1) | 5,489 | 97.4 (96.9–97.8) | |
| 35–44 years | 3,136 (18.9%) | 234 | 6.2 (5.4–7.2) | 2,902 | 93.8 (92.8–94.6) | |
| 45–54 years | 1,651 (10.1%) | 104 | 6.1 (4.9–7.5) | 1,547 | 93.9 (92.5–95.1) | |
| 55–64 years | 1,138 (5.8%) | 39 | 3.4 (2.4–4.7) | 1,099 | 96.6 (95.3–97.6) | |
| Total | 19,136 | 614 | 3.0 (2.8–3.3) | 18,522 | 97 (96.7–97.2) | |
| **Marital status** | | | | | | |
| Never married | 7,103 (35.5%) | 71 | 1.0 (0.7–1.3) | 7,032 | 99.0 (98.7–99.3) | 0.0001 |
| Married or living together | 9,418 (52.1%) | 285 | 2.8 (2.4–3.2) | 9,133 | 97.2 (96.8–97.6) | |
| Divorced or separated | 1,723 (8.5%) | 144 | 7.7 (6.5–9.3) | 1,579 | 92.3 (90.7–93.5) | |
| Widowed | 7,72 (3.9%) | 112 | 14.7 (12.1–17.7) | 660 | 85.3 (82.3–87.9) | |
| Total | 19,016 | 612 | 3.0 (2.8–3.3) | 18,404 | 97 (96.7–97.2) | |
| **Education level** | | | | | | |
| No education | 2,004 (11.9%) | 121 | 5.2 (4.2–6.3) | 2,279 | 94.8 (93.7–95.8) | <0.0001 |
| Primary | 6,803 (35.5%) | 291 | 4.2 (3.7–4.8) | 6,512 | 95.8 (95.2–96.3) | |
| Secondary | 5,488 (28.7%) | 141 | 2.4 (2.0–2.9) | 5,347 | 97.6 (97.1–98.0) | |
| More than secondary | 4,376 (24%) | 58 | 1.0 (0.7–1.4) | 4,318 | 99.0 (98.6–99.3) | |
| Total | 19,067 | 611 | 3.0 (2.8–3.3) | 18,456 | 97 (96.7–97.2) | |
| **Employment status in the last 12 months** | | | | | | |
| Did not worked | 313 (49.4) | 313 | 2.8 (2.5–3.2) | 10642 | 97.2 (96.8–97.5) | |
| Worked 12 last months | 298 (50.6) | 298 | 3.2 (2.8–3.7) | 7,856 | 96.8 (96.3–97.2) | 0.158 |
| Total | 19,109 | 611 | 3.0 (2.8–3.3) | 18,498 | 97.0 (96.7–97.2) | |
| **Number of sexual partners in the past 12-months** | | | | | | |

(*Continued*)

**Table 1.** (Continued)

| Characteristics | N (Weighted %) | HIV Infection Status | | | | P-value |
|---|---|---|---|---|---|---|
| | | Positive | | Negative | | |
| | | n | Weighted % (95% CI) | n | Weighted % (95% CI) | |
| None | 3,689 (27.7) | 241 | 6.0 (5.2–6.9) | 3,448 | 94.0 (93.1–94.8) | <0.0001 |
| One partner | 8,778 (68.2) | 270 | 2.9 (2.5–3.3) | 8,508 | 97.1 (96.7–97.5) | |
| Two or more partners | 497(4.1) | 25 | 3.1 (2.0–5.0) | 472 | 96.9 (95.0–98.0) | |
| Total | 12,964 | 536 | 3.8 (3.4–4.1) | 12,428 | 96.2 (95.9–96.6) | |
| **Gender of household head** | | | | | | |
| Male headed HH | 9,343 (53.7%) | 213 | 2.2 (1.9–2.6) | 9,130 | 97.8 (97.4–98.1) | <0.0001 |
| Female Headed Households | 9,793 (46.3%) | 401 | 4.0 (3.6–4.5) | 9,392 | 96.0 (95.5–96.4) | |
| Total | 19,136 | 614 | 3.0 (2.8–3.3) | 18,522 | 97.0 (96.7–97.2) | |
| **Food insecurity in the past 4 weeks** | | | | | | |
| No | 18,223 (95.5%) | 552 | 2.9 (2.7–3.2) | 17,671 | 97.1 (96.8–97.3) | <0.0001 |
| Yes | 808 (4.5%) | 56 | 5.0 (3.7–6.8) | 752 | 95.0 (93.2–96.3) | |
| Total | 19,031 | 608 | 3.0 (2.8–3.3) | 18,423 | 97.0 (96.7–97.2) | |
| **First sex before age 15** | | | | | | |
| First sex 15 + years | 17,735 (95.0%) | 536 | 2.9 (2.6–3.1) | 17,199 | 97.1 (96.9–97.4) | <0.0001 |
| First sex before 15 | 1,014 (5.0%) | 70 | 6.8 (5.3–8.7) | 944 | 93.2 (91.3–94.7) | |
| Total | 18,749 | 606 | | 18,143 | | |
| **Condom use at last sex in the past 12 months** | | | | | | |
| Used | 793 (6.4%) | 83 | 9.6 (7.5–12.2) | 710 | 90.4 (87.8–92.5) | <0.0001 |
| Did not use | 8,192 (65.2%) | 203 | 2.3 (2.0–2.7) | 7,989 | 97.7 (97.3–98.0) | |
| No sex | 3,689 (28.4%) | 241 | 6.0 (5.2–6.9) | 3,448 | 94.0 (93.1–94.8) | |
| Total | 12,674 | 527 | 3.1 (2.8–3.3) | 12,147 | 96.9 (96.7–97.2) | |
| **Ever tested for HIV** | | | | | | |
| Never tested | 5740 (29.5%) | 51 | 1.0 (0.7–1.3) | 5,689 | 99.0 (98.7–99.3) | |
| Ever tested | 13.186 (70.5%) | 561 | 3.9 (3.6–4.3) | 12,625 | 96.1 (95.7–96.4) | <0.0001 |
| **Total (N)** | **19,136** | 614 | **3.0 (2.8–3.3)** | **18,522** | **97.0 (96.7–97.2)** | |

had no sexual partner in the past 12 months (6.0% [95% CI: 5.2–6.9]), and those who used condom in the past 12 months (9.6% [95% CI: 7.5–12.2]). The differences in HIV prevalence across the various strata of the above variables were statistically significant (p<0.0001).

## Progress towards UNAIDS 90-90-90 targets and associated factors

Data on awareness about HIV positive status, ART (based on ARV drugs testing), and achievement of VL suppression was available for 609 (99.2%) of the 614 HIV-positive participants. In terms of the unconditional UNAIDS targets, 79% were aware of their HIV status, 77% of were on ART and 67% had suppressed viral loads (Fig 2). Among those aged 15–24 years, VLS achievement was much lower (46.8%).

Details of Ethiopia's progress in achieving each of the cascade UNAIDS 90-90-90 targets by 2018 are summarized in Table 2.

**Awareness status (first 90).** Among HIV-positive adults (79.0% [95% CI: 74.7–82.7]) were aware of their positive status when adjusted for ARV detection. Weighted awareness of HIV-positive status was high for the Benishangul Gumuz and Tigray regions (94.2% [95% CI: 68.3–99.2]) and (89.5% [95% CI: 74.7–96.1]), respectively, and the lowest, (66.6% [95% CI: 50.9–79.3]) for Gambella. There was a significant difference by gender (83.3% [95% CI 79.0–

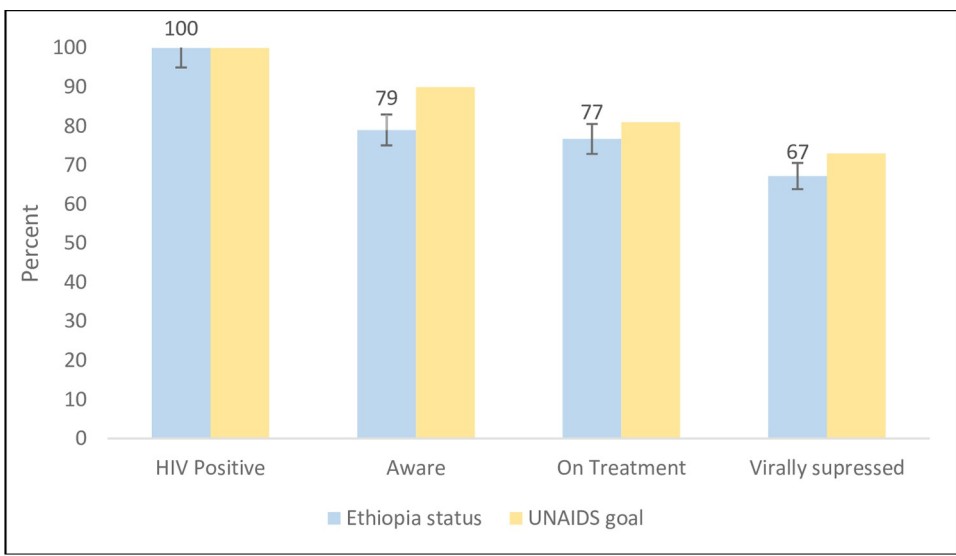

**Fig 2. Achievements of unconditional UNAIDS 90-90-90 targets among participants aged 15–64 years in urban Ethiopia, Ethiopia population-based HIV impact assessment, 2017–2018 (n = 609).**

86.9]) in females vs (70.0% [95% CI: 0.6–78.1]) in males and by gender of the HH head (83.6% [95% CI: 78.9–87.5]) female-headed vs (71.9% [95% CI: 63.8–78.7) male-headed.

**Treatment status (second 90).** Based on both self-reported ART status and detection of ARVs in their blood (97.1% [95% CI: 95.0–98.3]) of those aware of their status were on ART. Though there were some numerical differences in the association between the second 90 and administrative region, sex, and age, the differences were not statistically significant. The proportion of those on ART ranged from 92.0% (95% CI: 72.8–98.0) in Gambella to 100% in most of the regions and was above 95.4% across categories by sex, age, marital status, gender of head of HH, and condom use at the last sex in the past 12 months.

**VLS (third 90).** Among adults on ART (87.6% [95% CI: 83.9–90.5]) had VLS, ranging from (74.3% [95% CI: 54.9–87.3]) among the youth to (92.7% [95% CI: 77.7–97.9]) among older adults ages 50–64. Across all age groups, there was no statistically significant difference in VLS among females and males on ART. Though there were differences in the association between the third 90 and administrative region, and sex, and age, these did not achieve statistically significance. For VLS, the proportion ranged from 78.3% (95% CI: 60.8–89.3) in Tigray to 96.4% (95% CI: 78.1–99.5) in the Benishangul Gumuz region, similarly high in both sex groups, and low (74.3% [95% CI: 54.9–87.3]) in the age group 15–24 years. As shown in Fig 3, females performed better in the first and third 90 targets. Awareness was higher among those aged 25 years and older (>79.2%) compared to those15-24 years (63.0%) (Fig 4).

The results of a multivariate logistic regression analysis of the independent variables on HIV-positive status awareness and conditional estimates of having achieved VLS are given along with the adjusted odds ratios (aOR) in Table 3. Participants' sex, age, and condom use were independently and significantly associated with awareness about HIV-positive status. Females were more likely to be aware of their HIV-positive status (aOR = 2.8 [95% CI: 1.38–5.51]) as compared to males. The odds of being aware of HIV-positive-status was significantly higher with increasing age, (aOR = 4.2 [95% CI: 1.50–11.91]) for those 25 to 34-years. (aOR = 5.5 [95% CI: 2.03–15.01]) for those 35 to 44 years, (aOR = 7.0 [95% CI: 1.95–24.87]) for those 45 to 54 years, and (aOR = 11.4 [95% CI: 2.52–51.79]) for those 55 to 64 years compared to the youth 15–24 years of age. Participants who used condoms at last sex in the past 12

**Table 2. Association of socio-demographic and behavioural characteristics with the 90-90-90 targets among participants aged 15–64 years, Ethiopia Population-based HIV impact assessment, 2017–2018.**

| Characteristics | Awareness of HIV Positive Status | | On Antiretroviral Treatment | | Achieved Viral Suppression | |
|---|---|---|---|---|---|---|
| | n | Weighted % (95%CI) | n | Conditional weighted % (95%CI) | n | Conditional weighted % (95%CI) |
| **Region** | | | | | | |
| Tigray | 38 | 89.5 (74.7–96.1) | 34 | 100 | 34 | 78.3 (60.8–89.3) |
| Afar | 32 | 75.1 (57.1–87.2) | 24 | 100 | 24 | 86.4 (64.7–95.7) |
| Amhara | 118 | 85.2 (76.2–91.2) | 103 | 100 | 103 | 92.1 (84.5–96.2) |
| Oromia | 148 | 76.0 (67.2–83.0) | 119 | 95.2 (89.7–97.9) | 113 | 89.6 (82.7–94.0) |
| Somali | 8 | 100 | 8 | 100 | 8 | 90.0 (52.4–98.7) |
| Benishangul Gumuz | 20 | 94.2 (68.3–99.2) | 19 | 100 | 19 | 96.4 (78.1–99.5) |
| SNNPR | 49 | 73.3 (58.9–84.1) | 36 | 94.4 (80.0–98.6) | 34 | 85.2 (68.6–93.9) |
| Gambella | 44 | 66.6 (50.9–79.3) | 30 | 92.0 (72.8–98.0) | 28 | 88.4 (69.5–96.2) |
| Harari | 32 | 73.3 (55.5–85.7) | 23 | 100 | 23 | 88.1 (68.2–96.3) |
| Addis Ababa | 86 | 74.8 (64.0–83.2) | 65 | 95.2 (87.7–98.3) | 61 | 80.2 (68.2–88.4) |
| Dire Dawa | 34 | 83.9 (66.3–93.2) | 29 | 100 | 29 | 88.0 (71.6–95.5) |
| **Sex** | | | | | | |
| Female | 456 | 83.3 (79.0–86.9)* | 378 | 96.4 (93.5–98.0) | 366 | 86.1 (81.7–89.6) |
| Male | 153 | 70.0 (60.6–78.1) | 112 | 98.9 (94.1–99.8) | 110 | 91.1 (83.0–95.6) |
| **Age group** | | | | | | |
| 15–24 years | 62 | 63.0 (48.1–75.8) | 39 | 100 | 39 | 74.3 (54.9–87.3) |
| 25–34 years | 172 | 79.9 (70.8–86.7) | 141 | 95.8 (90.3–98.2) | 135 | 82.8 (74.0–89.1) |
| 35–44 years | 233 | 79.2 (71.9–85.1) | 190 | 96.7 (92.5–98.6) | 184 | 89.4 (83.5–93.3) |
| 45–54 years | 104 | 82.2 (72.4–89.1) | 87 | 98.9 (92.4–99.8) | 86 | 92.7 (84.7–96.7) |
| 55–64 years | 38 | 85.5 (69.2–94.0) | 33 | 95.8 (75.5–99.4) | 32 | 92.7 (77.7–97.9) |
| **Marital status** | | | | | | |
| Never married | 71 | 76.2 (63.0–85.8) | 53 | 97.8 (86.1–99.7) | 52 | 76.9 (62.4–87.0) |
| Married/living together | 285 | 76.9 (70.1–82.6) | 228 | 97.2 (93.6–98.8) | 221 | 92.5 (87.8–95.5) |
| Divorced or separated | 140 | 80.3 (71.1–87.2) | 114 | 97.7 (93.1–99.3) | 111 | 88.4 (80.1–93.5) |
| Widowed | 111 | 84.9 (75.5–91.1) | 94 | 95.8 (87.7–98.7) | 91 | 83.1 (72.9–90.0) |
| **Education** | | | | | | |
| No education | 120 | 86.2 (77.2–92.0) | 101 | 95.2 (87.9–98.2) | 96 | 83.6 (73.9–90.2) |
| Primary | 288 | 76.2 (69.5–81.9) | 230 | 98.3 (95.5–99.4) | 226 | 87.8 (82.2–91.8) |
| Secondary | 141 | 81.9 (73.2–88.2) | 117 | 95.7 (89.1–98.4) | 113 | 87.2 (78.6–92.7) |
| More than secondary | 57 | 71.3 (54.6–83.7) | 40 | 99.4 (95.5–99.9) | 39 | 98.8 (95.2–99.7) |
| **Gender of head of household** | | | | | | |
| Male headed HHs | 213 | 71.9 (63.8–78.7) | 161 | 97.8 (93.9–99.2) | 156 | 91.4 (85.4–95.0) |
| Female headed HHs | 396 | 83.6 (78.9–87.5)* | 329 | 96.7 (93.7–98.3) | 320 | 85.5 (80.5–89.3) |
| **Condom use at last sex in the past 12 months** | | | | | | |
| Used condom | 83 | 91.3 (82.2–96.0) | 75 | 98.6 (90.6–99.8) | 74 | 93.6 (84.7–97.5) |
| Did not use condom | 201 | 70.0 (61.4–77.3) | 150 | 95.4 (90.2–97.9) | 142 | 91.1 (84.7–95.0) |
| No sex, past 12 months | 238 | 82.2 (75.4–87.4) | 199 | 97.5 (93.4–99.1) | 195 | 86.0 (79.7–90.6) |
| Total | 609 | 79.0 (74.7–82.7) | 490 | 97.1 (95.0–98.3) | 476 | 87.6 (83.9–90.5) |

* Statistically significant (nonoverlapping of confidence intervals)

months were more likely to be aware of their HIV- positive status (aOR = 5.1 [95% CI: 1.68–15.25]), compared to those who did not use condom. VLS was strongly and significantly associated with increasing educational status; individuals with secondary education and above

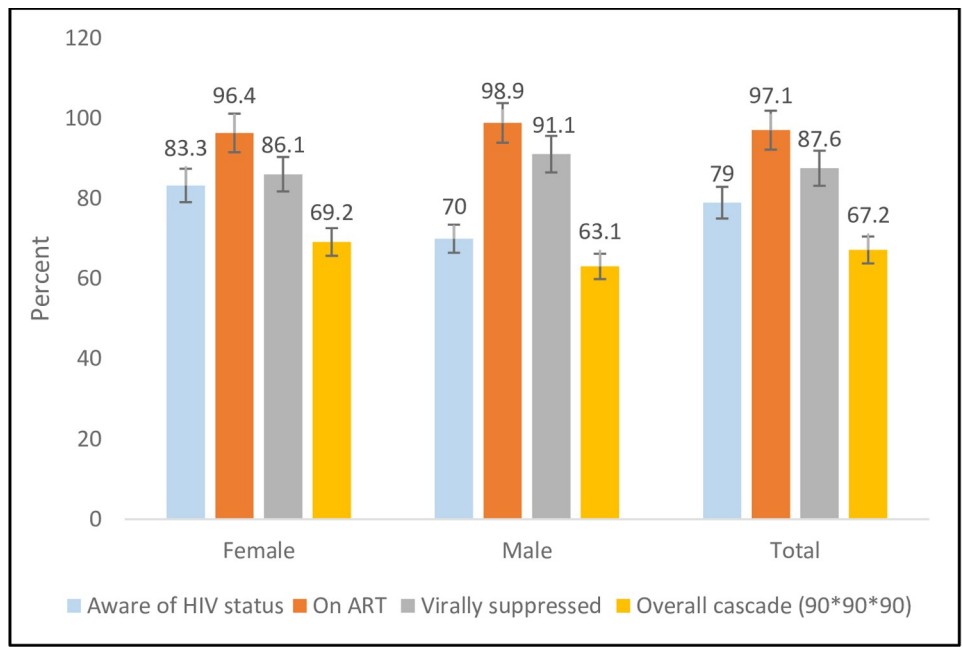

**Fig 3. ARV adjusted conditional 90-90-90 targets achieved among adult participants by sex, Ethiopia population-based HIV impact assessment, 2017–2018.**

were more likely to have achieved viral suppression (aOR = 8.2 [95% CI: 1.82–37.07]) compared with those with no education.

## Discussion

The HIV prevalence estimate from EPHIA was similar to urban HIV prevalence obtained from the EDHS in 2016 [12], which was 3.0%. Ethiopia achieved 67% of the expected overall

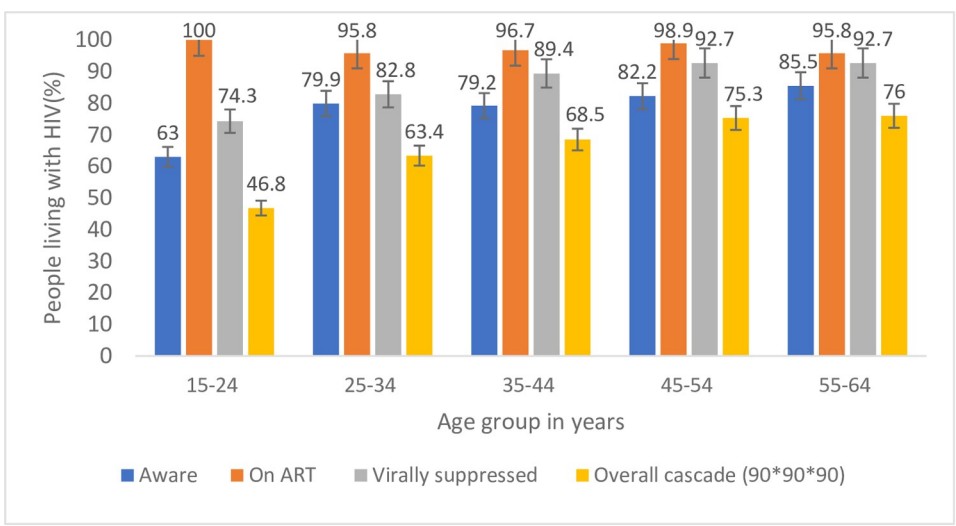

**Fig 4. ARV adjusted conditional 90-90-90 targets achieved among adult participants by age, Ethiopia population-based HIV impact assessment, 2017–2018.**

**Table 3. Factors associated with HIV positive awareness among people living with HIV and viral load suppression among HIV positive adult participants, Ethiopia population-based HIV impact assessment, 2017–2018\*.**

| Characteristics | Aware of HIV+ status | | Virally suppressed | |
|---|---|---|---|---|
| | among people living with HIV | | among HIV+ participants on ART | |
| | AOR (95%CI) | P-value | AOR (95%CI) | P-value |
| **Region** | | | | |
| Tigray | 2.4 (0.32–18.56) | 0.392 | 0.7 (0.17–3.12) | 0.660 |
| Afar | 1.0 (0.14–6.99) | 0.982 | 1.2 (0.23–6.85) | 0.801 |
| Amhara | 1.2 (0.18–8.26) | 0.833 | 3.4 (0.65–18.22) | 0.144 |
| Oromia | 0.7 (0.13–4.33) | 0.733 | 1.8 (0.37–8.35) | 0.478 |
| Benishangul Gumuz | 5.6 (0.61–51.67) | 0.128 | 2.8 (0.71–11.24) | 0.140 |
| SNNPR | 1.1 (0.14–8.0) | 0.960 | 0.8 (0.13–5.24) | 0.836 |
| Gambella | 1 | 1 | 1 | 1 |
| Harari | 0.4 (0.05–3.61) | 0.437 | 0.7 (0.14–3.56) | 0.671 |
| Addis Ababa | 0.5 (0.09–3.37) | 0.511 | 1.0 (0.22–4.32) | 0.961 |
| Dire Dawa | 1.0 (0.15–7.21) | 0.965 | 1.0 (0.17–6.31) | 0.975 |
| **Sex** | | | | |
| Female | 2.8\*\* (1.38–5.51) | 0.004 | 0.7 (0.15–3.43) | 0.680 |
| Male | 1 | 1 | 1 | 1 |
| **Age group** | | | | |
| 15–24 years | 1 | 1 | 1 | 1 |
| 25–34 years | 4.2\*\* (1.50–11.91) | 0.007 | 1.3 (0.14–11.90) | 0.816 |
| 35–44 years | 5.5\*\* (2.03–15.01) | 0.001 | 2.0 (0.22–18.23) | 0.545 |
| 45–54 years | 7.0\*\* (1.95–24.87) | 0.003 | 2.1 (0.20–21.49) | 0.536 |
| 55–64 years | 11.4\*\* (2.52–51.79) | 0.002 | 4.8 (0.12–192.89) | 0.404 |
| **Marital status** | | | | |
| Never married | 1 | 1 | 1 | 1 |
| Married or living together | 1.0 (0.35–2.56) | 0.923 | 3.4 (0.75–15.37) | 0.112 |
| Divorced or separated | 0.7 (0.26–1.97) | 0.510 | 3.9\* (0.80–18.83) | 0.092 |
| Widowed | 1.1 (0.35–3.15) | 0.922 | 2.2 (0.52–8.96) | 0.284 |
| **Gender of head of household** | | | | |
| Male headed HH | 1 | 1 | 1 | 1 |
| Female Headed Households | 1.2 (0.60–2.43) | 0.593 | 0.9 (0.29–2.78) | 0.841 |
| **Education level** | | | | |
| No education | 1 | 1 | 1 | 1 |
| Primary | 0.9 (0.44–1.86) | 0.792 | 1.5 (0.62–3.403 | 0.384 |
| Secondary | 1.4 (0.59–3.08) | 0.474 | 1.9 (0.58–6.10) | 0.288 |
| More than secondary | 0.8 (0.283–2.01) | 0.571 | 8.2\*\* (1.82–37.07) | 0.006 |
| **Condom use at last sex in the past 12 months** | | | | |
| Used condom | 5.1\*\* (1.68–15.25) | 0.004 | 1.4 (0.36–5.77) | 0.605 |
| Did not use condom | 1 | 1 | 1 | 1 |
| No sex in the past 12 month | 1.5 (0.70–3.0) | 0.314 | 0.8 (0.37–1.89) | 0.656 |

\* Antiretroviral treatment (ART) not included here as achievement is more than 98%

\*\* Statistically significant (p<0.05)

UNAIDS target of 73% for the three targets. Achievements in the second and third 90s indicate that the country made considerable progress towards these targets. However, the gap in achievement was wider (11%) for the first 90 target. Based on the EPHIA HIV prevalence estimate and Ethiopia's urban population size projected from the 2007 census for 2018 [15], it was

estimated that there were approximately 384,000 adults living with HIV in urban Ethiopia. Approximately 303,000 of them were aware of their HIV-positive status, 295,000 were on treatment, and 258,000 achieved VLS.

Among HIV-positive adults in EPHIA, 79% knew that they were HIV positive. This was similar to the awareness level estimated by another study in Ethiopia [20], but lower than the 2020 UNAIDS global estimate for eastern and southern Africa [4], which reported an awareness level of 87%. The PHIA surveys in several other countries [21] as well as other small-scale studies in Africa [22, 23] showed that countries in the continent were at various stages of progress towards achieving the first 90 target. Ethiopia was among those which performed relatively better [24], but still had gaps in achieving the first 90 or awareness target.

Females were more likely to report that they knew their HIV-positive status than their male counterparts, which was consistent with findings from other studies [25–28]. Like what was reported by others from other sub-Saharan Africa (SSA) countries [28–31], we found that the PLHIV in the age group 15–24 were less likely to be aware of their HIV status. The challenge of reaching the youth in SSA for HIV testing was identified by some other studies as well [32]. As the youth constitutes a substantial segment of the population in Ethiopia [13], focusing on this group by the national programme can help to consolidate and improve the gains in HIV testing and, consequently, achieve treatment and VLS targets. Concurring with a report from Mozambique [25], awareness of HIV-positive status was also associated with condom use in the past 12 months in urban Ethiopia. This emphasizes the implications of high-risk sexual behaviours such as non-condom use and the associated unawareness of HIV-positive status would have for HIV programming.

Given the low HIV prevalence and that the epidemic is concentrated in urban areas and as emphasized by others [33–35], Ethiopia needs to strengthen targeted testing by focusing on subgroups at higher risk based on behavioural, clinical, or demographic characteristics, including males, the youth, those with no formal education, and key and priority populations. Based on results from the EPHIA survey and other reports [36, 37], Ethiopia would need to address the considerable variability in awareness by administrative region, gender, and other social and behavioural determinants to enhance the enrolment of PLHIV in care and treatment services.

Ethiopia's performance in the second 90, though remarkable, was still short of achieving the UNAIDS target by 2020. Moreover, progress with this target may change as efforts to reach the first 90 target continue and ART coverage among those aware must be sustained. The UNAIDS global report 2019 indicated that, among people who knew their HIV status, 78% were accessing treatment [38]. The achievement varied by country [39] and Ethiopia was among those which performed well.

Among HIV-positive adults who knew their HIV-positive status and were on treatment, 87.6% were virally suppressed, indicating that Ethiopia was close to achieving the 2020 VLS target. Indeed, this achievement was higher than the global average of 78%. However, VLS was as low as 74.3% among the youth posing a challenge that would need attention by the national HIV programme. VLS among adult men also remained behind, requiring due focus as two-thirds of all HIV transmissions were reported to be from adult men to women [40]. The importance of engaging the youth and adult men in care and treatment to ensure that they can maintain undetectable viral load has also been emphasized by other reports [32].

There is a need for giving focus to increasing awareness in the context of Ethiopia as most of the PLHIV who were not virally suppressed in EPHIA were unaware of their status. Unlike reports from other studies [41–43], age and gender were not significantly associated with VLS in the EPHIA survey. Interventions seeking to improve coverage of HIV testing, adherence,

and retention in care to achieve VLS may require tailored programme components and strategies to increase awareness with an emphasis on the young population and men.

The EPHIA results and data from other studies clearly indicate that Ethiopia should strengthen the HIV case identification and treatment programmes. Moreover, there is a consensus that without achieving the overarching target of the first 90, the control of the HIV epidemic will not be achieved [44]. Indeed, Ethiopia has recently taken the initiative of establishing an HIV case-based surveillance system to capture recent infections among newly diagnosed PLHIV and ensure that they receive the continuum of HIV care and treatment services and monitor individual and cluster-level outcomes as well as the country's progress towards achieving the 95-95-95 targets.

As emphasized by other reports [45, 46], it is essential to strengthen the health system to maintain the continued expansion of HIV services. Progress with the updated UNAIDS 95-95-95 targets is also being challenged by the COVID-19 pandemic [47], which has negatively impacted HIV services including testing and access to ART; a negative impact that is likely to continue even after the pandemic is controlled [48, 49]. It is suggested that the country, as in other similar settings [50],

needs to step up its efforts to meet the challenges posed by the dual emergencies of HIV and COVID 19 epidemics.

## Limitations

It should be noted that the EPHIA survey did not include the rural population, which could differ from the urban population with respect to HIV diagnosis, treatment, and VLS as well as participant characteristics and health facility readiness [51]. However, this could not be a major limitation as the survey covered the urban areas, where the epidemic in Ethiopia is concentrated. Some 6.4% of the eligible participants declined the interview and 5.1% eligible for laboratory specimens sampling declined blood draw. This could have introduced some bias, but this would not be substantial as the response rate was still much higher than the expected rate. The Somali region had a relatively small number of HIV-positive individuals and was excluded from the multivariate analysis. The survey did not include key populations such as commercial sex workers, which are high-risk groups considered as the major divers of the epidemic. The survey also had the inherent limitation of a cross-sectional study design, which does not allow for assessing causal relationships.

## Conclusion

EPHIA provided scientifically sound evidence on the prevalence and status of UNAIDS 90-90-90 targets in urban Ethiopia. Our analysis showed that Ethiopia was on the verge of achieving the second and third 90s, but lagged behind in the first 90 target, and identified factors that negatively affected the program from achieving the targets. The national HIV care and treatment programme needs to target population groups with low access to HIV services such as men, the youth, and those with no education to improve awareness, treatment, and viral load suppression in urban Ethiopia. Achieving the first 90 target would require a continued effort to scale up HIV testing through innovative approaches such as self-testing and outreach community-friendly testing services with a special focus on high HIV burden geographic settings, the youth, men, and other high social risk groups living with HIV. Further exploratory research is needed to understand the role of the individual, community, and structural level barriers to achieving the updated interim 95-95-95 targets and identify the magnitude and determinants of outcome among the high-risk social groups.

## The EPHIA Study Team

| | |
|---|---|
| Yimam Getaneh (PI)<br>Saro Abdella<br>Wudinesh Belete<br>Tsigereda Kifle<br>Abebe Habteselassie<br>Minilik Demissie<br>G/medhin G/Michael<br>Habtamu Teklie<br>Ebba Abate | Ethiopian Public Health Institute |
| Eleni Seyoum | Federal HIV/AIDS Prevention and Control Office |
| Esayas Muleta | Central Statistics Agency |
| Frehywot Eshetu<br>Jelaludin Ahmed<br>Clare Dykewicz<br>Ashenafi Haile<br>Yared Tedla<br>Jeff Hanson<br>Christine Ross<br>Biniyam Eskinder<br>Solomon Fekadie<br>Wondimu Teferi | U.S Centers for Disease Control and Prevention, Ethiopia |
| Drew Voetsch (PI)<br>Aderonke S. Ajiboye<br>Sehin Birhanu<br>Kristin Brown<br>Edith Nyangoma<br>Bharat Parekh<br>Hetal Pate<br>Christine W. West | U.S Centers for Disease Control and Prevention, Atlanta |
| Sileshi Lulseged (PI)<br>Zenebe Melaku<br>Halegnaw Eshete<br>Terefe Gelibo<br>Belete Tegbaru<br>Yohanes Demissie<br>Nadew Tademe | ICAP-Columbia University, Ethiopia |
| Jessica Justman (PI)<br>David Hoos<br>Mansoor Farahani<br>Karampreet Sachathep<br>Suzue Saito<br>Andrea Low<br>Chelsea Solmo | ICAP-Columbia University-New York |

## Acknowledgments

We would like to extend our thanks to the leadership at the Ministry of Health, EPHI, the RHBs sub-regional units, CDC and ICAP leadership for their administrative support in organizing and conducting the survey. Our thanks also go to field coordinators, supervisors, and data collectors for their dedicated work and all study participants for generously providing the necessary information.

This project was conducted using the U.S. President's Emergency Plan for AIDS Relief (PEPFAR) funds obtained though the U.S Centre for Disease Control and Prevention (CDC) under the term of cooperative agreement #U2GGH001226. The findings and conclusions in this report are those of the authors and do not necessarily represent the official position of the funding agency.

## Author Contributions

**Conceptualization:** Sileshi Lulseged, Zenebe Melaku, Abebe Habteselassie, Christine A. West, Terefe Gelibo, Wudinesh Belete, Mansoor Farahani, Minilik Demissie, Christine E. Ross.

**Data curation:** Sileshi Lulseged, Zenebe Melaku, Christine A. West, Terefe Gelibo, Mansoor Farahani.

**Formal analysis:** Sileshi Lulseged, Christine A. West, Terefe Gelibo, Mansoor Farahani.

**Investigation:** Sileshi Lulseged, Zenebe Melaku, Abebe Habteselassie, Terefe Gelibo, Wudinesh Belete, Fana Tefera, Mansoor Farahani, Minilik Demissie, Wondimu Teferi, Saro Abdella, Sehin Birhanu, Christine E. Ross.

**Methodology:** Sileshi Lulseged, Zenebe Melaku, Abebe Habteselassie, Christine A. West, Terefe Gelibo, Wudinesh Belete, Fana Tefera, Mansoor Farahani, Minilik Demissie, Wondimu Teferi, Saro Abdella, Sehin Birhanu, Christine E. Ross.

**Project administration:** Sileshi Lulseged, Zenebe Melaku, Christine E. Ross.

**Resources:** Sileshi Lulseged, Zenebe Melaku.

**Supervision:** Sileshi Lulseged, Abebe Habteselassie, Terefe Gelibo, Wudinesh Belete, Fana Tefera, Mansoor Farahani, Minilik Demissie, Wondimu Teferi, Saro Abdella, Sehin Birhanu, Christine E. Ross.

**Validation:** Sileshi Lulseged, Zenebe Melaku, Abebe Habteselassie, Christine A. West, Terefe Gelibo, Wudinesh Belete, Fana Tefera, Mansoor Farahani, Minilik Demissie, Wondimu Teferi, Saro Abdella, Sehin Birhanu, Christine E. Ross.

**Visualization:** Sileshi Lulseged, Zenebe Melaku, Abebe Habteselassie, Christine A. West, Terefe Gelibo, Wudinesh Belete, Fana Tefera, Mansoor Farahani, Minilik Demissie, Wondimu Teferi, Saro Abdella, Sehin Birhanu, Christine E. Ross.

**Writing – original draft:** Sileshi Lulseged, Zenebe Melaku, Abebe Habteselassie, Christine A. West, Terefe Gelibo, Wudinesh Belete, Fana Tefera, Mansoor Farahani, Minilik Demissie, Wondimu Teferi, Saro Abdella, Sehin Birhanu, Christine E. Ross.

**Writing – review & editing:** Sileshi Lulseged, Zenebe Melaku, Abebe Habteselassie, Christine A. West, Terefe Gelibo, Wudinesh Belete, Fana Tefera, Mansoor Farahani, Minilik Demissie, Wondimu Teferi, Saro Abdella, Sehin Birhanu, Christine E. Ross.

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
