## [Decision Letter · Decision Letter 0]

3 Nov 2021

PONE-D-21-26375Progress towards controlling the HIV epidemic in urban Ethiopia: Findings from the 2017-2018 Ethiopia population-based HIV impact assessment surveyPLOS ONE

Dear Dr. Lulseged,

Thank you for submitting your manuscript to PLOS ONE. After careful consideration, we feel that it has merit but does not fully meet PLOS ONE’s publication criteria as it currently stands. Therefore, we invite you to submit a revised version of the manuscript that addresses the points raised during the review process.

Both reviewers agree on the significant contribution this work will make on HIV policy.  However, they have also both identified significant issues that must be addressed in order for this work to advance at PLOS ONE.  Importantly, I note that Reviewer #1 answered "No" to "Has the statistical analysis been performed appropriately and rigorously?" and Reviewer #2 answered "Partly" and "No" to the questions "Is the manuscript technically sound, and do the data support the conclusions?" and "Is the manuscript presented in an intelligible fashion and written in standard English?", respectively. At PLOS ONE, a "Yes" to each of these items is required. While the authors must address all reviewer comments, I would like to draw special attention to the following issues.  Both reviewers found significant issues with grammar, sentence structure and typos and feel the manuscript requires a thorough copy-edit to improve clarity.  Reviewer #1 raises important concerns about the need for statistical analyses that include the simultaneous effects of multiple variables. The reviewers have both commented on the fact that the surveys focused on urban areas. While I agree that this may not be a major limitation (as Reviewer #1 feels), I do agree with Reviewer #2 that this fact should be introduced clearly and earlier in the manuscript so the reader can make their own assessment of this aspect.

We look forward to receiving your revised manuscript.

Kind regards,

David Gerberry, Ph.D.

Academic Editor

PLOS ONE

Journal Requirements:

2. In the Methods section of your manuscript, please provide additional information regarding the criteria used to determine emancipation of minors. Furthermore, please could you clarify for minors ages between 15-18, who were not considered emancipated , whether you obtained consent from parents or guardians of the minors included in the study or whether the research ethics committee or IRB specifically waived the need for their consent

3. Please include additional information regarding the survey or questionnaire used in the study and ensure that you have provided sufficient details that others could replicate the analyses. For instance, if you developed a questionnaire as part of this study and it is not under a copyright more restrictive than CC-BY, please include a copy, in both the original language and English, as Supporting Information

4. Please update your submission to use the PLOS LaTeX template. The template and more information on our requirements for LaTeX submissions can be found at http://journals.plos.org/plosone/s/latex

5. We note that Figure(s) Appendix 1, 2 and 3 in your submission contain [map/satellite] images which may be copyrighted. All PLOS content is published under the Creative Commons Attribution License (CC BY 4.0), which means that the manuscript, images, and Supporting Information files will be freely available online, and any third party is permitted to access, download, copy, distribute, and use these materials in any way, even commercially, with proper attribution. For these reasons, we cannot publish previously copyrighted maps or satellite images created using proprietary data, such as Google software (Google Maps, Street View, and Earth). For more information, see our copyright guidelines: http://journals.plos.org/plosone/s/licenses-and-copyright.

1. You may seek permission from the original copyright holder of Figure Appendix 1, 2 and 3 to publish the content specifically under the CC BY 4.0 license.  

Reviewers' comments:

Reviewer's Responses to Questions

**Comments to the Author**

1. Is the manuscript technically sound, and do the data support the conclusions?

Reviewer #1: Yes

Reviewer #2: Partly

2. Has the statistical analysis been performed appropriately and rigorously? 

Reviewer #1: No

Reviewer #2: Yes

3. Have the authors made all data underlying the findings in their manuscript fully available?

Reviewer #1: Yes

Reviewer #2: Yes

4. Is the manuscript presented in an intelligible fashion and written in standard English?

Reviewer #1: Yes

Reviewer #2: No

5. Review Comments to the Author

Reviewer #1: The authors analyzed data from the Ethiopian Population HIV Impact Assessment conducted in 2017/18 to assess the progress made towards attaining the UNAIDS- 90-90-90 targets and the associated socio-demographic and behavioural factors in Urban Ethiopia. They found that 79.0% of the respondents infected by HIV were aware of their HIV status based on self-report or detection of ARV metabolites; 97.1% of those aware, were on antiretroviral therapy; and of those on treatment, 87.6% had achieved viral suppression. Some disparities by population subgroups were found in achieving the first 90, while no disparities were found for the second and third 90 targets.

The authors’ findings are informative and will guide program implementation efforts in urban areas in Ethiopia and in other countries in Sub Saharan Africa.

I have the following general comments:

1. There are some typos in the abstract and in the main document, the authors may consider reading through carefully and correct them including sentence structure. I have highlighted some examples here below.

2. Analysis for the associated socio-demographic and behavioural factors have been limited to bivariate level, hence no consideration of simultaneous effect of multiple variables – which is a more realistic representation of our environments, the conclusions drawn here may not therefore be realistic to the real life situation. Additionally, the authors have previously published a paper “Factors associated with unawareness of HIV positive status in urban Ethiopia ….” in which the correct analysis approach was followed.

3. More detailed explanation of the survey design; field processes such as consenting of survey participants; and ethical considerations are already published in the survey report. The authors could focus more on analysis methods for the first and second objectives and in discussing the implication of the analysis results to program implementation.

4. In the appendices, figures/maps are presented but there is no description of analysis methods used and linkage to the objectives of the study.

5. Limitations, please add more. Focusing the survey in urban areas, I think may not qualify as a major limitation. The authors have rightly noted in the conclusions, these are the most affected sub areas which require more intensified efforts in the absence of adequate resources.

6. Methods of analysis is not explicitly stated in the methods section of the abstract. This will be helpful to link the results to the methods and the conclusions drawn.

Additional specific comments

- Line 63: “… however, with the end of 2020, ….” Review and revise the statement

- Review and restate correctly the statement of targets. It does not read correct --Line 67: should read “…. 95% of PLHIV who know their status are on treatment”

- Line 68: “…..have virally suppressed” replace with “…. have suppressed viral loads”

- Line 181-182 “…62% did not use a condom the last sex …” change to “… 62% did not use a condom at last sex …”

- In Table 1 “ first sex bonfire age 15” should read “First sex before age 15 years”

- Line 205. “…among those aged 15-24…” should be “…. Among those aged 15-24 years…”

- Repetition - line 195 and Line 196 (“…..had sex before age 15 (6.8%)…)

- Line 309 “….receive….” is that supposed to be “….achieve...”

- Table 2: missing “n” for viral suppression for the subgroup “did not use a condom”

- Line 253: Is figure 5 (Appendix 1), prepared as part of this analysis or is reference? In either case, it needs to be clearly described in the methods section.

Reviewer #2: The authors report findings of a large household survey conducted in urban areas of Ethiopia. The survey collected socio-demographic and behavioral indicators as well as self-reported HIV and treatment status information via questionnaires, and assessed HIV status, antiretroviral treatment (ART) use, and viral load suppression empirically based on blood samples from consenting participants. The authors report estimates of HIV prevalence, as well as proportions of people living with HIV who know their status, are on ART, and/or who are virally suppressed. These methods are consistent with other Population-based HIV Impact Assessment surveys done in sub-Saharan Africa, are sound, and provide data appropriate for these analyses.

This manuscript is an important contribution to the research literature, as well as for understanding the HIV epidemic in urban areas of Ethiopia. I believe the manuscript itself would benefit from additional detail around the methods, and from copy-editing to improve clarity. I have discussed these in specific comments below.

Major comments:

1. I would like to see more care that these results are not presented as nationally representative, e.g. at lines 32 and 47-49, given the urban scope of the survey. I appreciate that the urban scope is discussed as a limitation; please include the rationale for this choice in the Methods section as well. Line 88-89 (“a reliable nationally representative number of PLHIV for HIV programmatic planning was lacking”) draws particular attention to this limitation, so it would help to address it earlier.

2. The role of the weights in producing these estimates is not clear (e.g., line 141). Please provide a bit more detail about the survey design and what these weights account for.

3. There is a big drop-off from the 25,416 selected household members to the 21,560 considered eligible (Figure 1). Please elaborate on the eligibility criteria a bit more, and the potential limitations those might impose. For example, how many potential respondents were ineligible because they did not speak one of the survey languages (line 155), and might their exclusion have biased estimates?

4. In the results section and especially the figures and tables, please make sure the denominators are clear when looking at the “absolute” cascade (percentages of all PLHIV) vs. the “conditional” cascade (e.g. viral suppression among PLHIV on ART).

5. There are a couple places where it seems overly strong to claim that there are no differences in estimates, when differences in estimates apparently exist but perhaps did not reach statistical significance. For example, at line 228 the authors report that the “second 90 does not differ by age, gender, or administrative region”, but there are some numerical differences in these estimates in Table 2. This also applies to the third 90 at line 234-235.

6. The 97% ART coverage among PLHIV who know their status is praiseworthy, but I think it would be appropriate to temper the claim that the second 90 target has been achieved (Line 278). That situation may change as efforts to reach the first 90 target continue. The first two 90’s together would imply that at least 81% of PLHIV are on ART, in comparison to the 77% reported in Figure 1, which owes to the gap in achievement of the first 90. Efforts to maintain high ART coverage among those aware must be sustained as HIV status awareness improves.

Minor comments:

1. Line 27: Missing punctuation before “some believed the targets were not achievable”?

2. Line 29-31: Perhaps the sentence starting “The overall cascade of HIV-positive people […]” should be in the Methods section.

3. Line 37: For context, it would help to list which ARVs were screened for in the methods section, and how those lined up with ARVs commonly in-use in Ethiopia when the survey was conducted.

4. Line 40-46: Please report uncertainty bounds about the estimates.

5. Line 42: “ARV” has not been defined.

6. Line 44-46: Since the 90-90-90 targets have not been formally defined in the preceding text, “variation in the achievement of the first 90 target” may be unclear to some readers. Please either define the three components of the 90-90-90 targets or describe this more directly ("variation in knowledge of HIV-positive status").

7. Line 55: “United Nations Programme of HIV/AIDS” should be “the Joint United Nations Programme on HIV/AIDS” as at line 25.

8. Line 71-82: Please check the references in this section. For example, reference (8) pertains to TB and does not seem to support the claims here regarding declines in HIV incidence and mortality.

9. Line 81-82: The sentence starting “Treatment failure[…]” seems a bit disconnected from the rest of the paragraph.

10. Line 96: “countries” should be “country’s”

11. Line 114: Please clarify whether these eleven regions refer to the nine regional states and two city administrations?

12. Line 139: Please close or remove the unmatched parentheses here and throughout the text and tables.

13. Line 170: Please define “EA”.

14. Table 1: (a) There seems to be a typo for the total HIV-negative by sex (n=46205). (b) Should “First sex bonfire age 15” be “First sex before age 15”?

15. Figure 2: (a) There should not be confidence intervals on the UNAIDS targets. (b) Please lighten the bar color for the survey-based estimates, as the lower limits of the uncertainty bounds are barely visible.

16. Table 2: The sample size for viral suppression among those who did not use a condom is missing.

17. Line 231: Should “condom use in the last 24 hours” be “condom use at last sex in the last 12 months”?

18. Figures 5-7: I believe these estimates come from different time points? Please clarify.

19. Line 287-88: “[…]two-thirds of all HIV transmissions were estimated to be from adult men to women.” Please provide a citation for this.

20. I spotted a couple issues in the references, please review: (a) Line 354: The title is incorrect. (b) Line 416: The author (UNAIDS) is garbled.

6. PLOS authors have the option to publish the peer review history of their article (what does this mean?). If published, this will include your full peer review and any attached files.

Reviewer #1: No

Reviewer #2: **Yes: **Robert Glaubius

---

## [Decision Letter · Decision Letter 1]

21 Jan 2022

PONE-D-21-26375R1Progress towards controlling the HIV epidemic in urban Ethiopia: Findings from the 2017-2018 Ethiopia population-based HIV impact assessment surveyPLOS ONE

Dear Dr. Lulseged,

Thank you for submitting your manuscript to PLOS ONE. After careful consideration, we feel that it has merit but does not yet fully meet PLOS ONE’s publication criteria as it currently stands. Therefore, we invite you to submit a revised version of the manuscript that addresses the points raised during the review process.

 Both reviewers have responded favorably to the changes made in Revision #1.  The article seems on its way toward publication in PLOS ONE.  All that remains is to address the remaining concerns of Reviewer #2. Provided that these primarily straightforward concerns are addressed in the next version, I do not anticipate needing to send the article back out to the reviewers.

We look forward to receiving your revised manuscript.

Kind regards,

David Gerberry, Ph.D.

Academic Editor

PLOS ONE

Journal Requirements:

Reviewers' comments:

Reviewer's Responses to Questions

**Comments to the Author**

1. If the authors have adequately addressed your comments raised in a previous round of review and you feel that this manuscript is now acceptable for publication, you may indicate that here to bypass the “Comments to the Author” section, enter your conflict of interest statement in the “Confidential to Editor” section, and submit your "Accept" recommendation.

Reviewer #1: All comments have been addressed

Reviewer #2: (No Response)

2. Is the manuscript technically sound, and do the data support the conclusions?

Reviewer #1: Yes

Reviewer #2: Yes

3. Has the statistical analysis been performed appropriately and rigorously? 

Reviewer #1: Yes

Reviewer #2: I Don't Know

4. Have the authors made all data underlying the findings in their manuscript fully available?

Reviewer #1: Yes

Reviewer #2: Yes

5. Is the manuscript presented in an intelligible fashion and written in standard English?

Reviewer #1: Yes

Reviewer #2: Yes

6. Review Comments to the Author

Reviewer #1: The Authors have now applied a more appropriate statistical analysis methodology, presented and explained the results well to my satisfaction. The explanations are now clearer and easy to follow.

Reviewer #2: The authors have largely addressed each of my comments. Most of my comments below are on copy-editing issues rather than methodological concerns, though I do have some questions regarding the new multivariate regression analysis.

1. I appreciate that the authors are now more explicit about "unconditional" and "conditional" estimates and targets. These terms should be defined in the methods section, or replaced with more direct statements, as has been done in Table 3 (e.g. "Virally suppressed among HIV+ participants on ART").

2. I also appreciate the new multivariate regression analysis. I answered "I don't know" with regards to the appropriateness and rigor of the approach because I feel we need a bit more information here. Could you please explain how you chose which explanatory variables to include? Was age modeled as a continuous variable or a categorical variable? Either is valid, but if modeled categorically, that might have diluted any age effects so it seems worth mentioning. Finally, Table 3 includes education level but Table 2 does not. Could that be added to Table 2 to provide more context?

3. Please review Table 1 carefully for accuracy and clarity:

3.a. Total rows for administrative region and sex incorrectly state the 95% CI for HIV prevalence as ".8-3.3"; this is reported as "2.8-3.3" at line 218.

3.b. Under "First sex before age 15", percentages HIV-positive and HIV-negative are missing.

3.c. HIV prevalence among those never tested cannot be right, "1 9 (0.7-1.3)", if 99.0% of those never tested were HIV-negative.

3.d. The 95% CI for the % of never-tested who were HIV-negative seems far too wide (8.7-99.0).

3.e. There are several typos, like square brackets sometimes used in place of parentheses, duplicate parentheses ("))"), and inconsistent use of commas and percentage signs. These may be minor individually, but together suggest that there could be other undetected errors in the results too.

4. The authors have improved many of the copy-editing issues identified in the previous round, but there are some issues remaining:

4.a. Line 42: "Awareness Females about HIV-positive[...]" perhaps should be "Awareness of HIV-positive status[...]"?

4.b. Lines 166-168 repeat lines 97-99; these could be merged.

4.c. Line 194-201: The paragraph duplicates the number eligible among household members identified (lines 192 & 195) and interviewed among eligible (lines 195 & 199). Perhaps this paragraph could be revised to eliminate redundancy?

4.d. Line 207: "[...]34.9% were 1 in the[...]". Please remove "1".

4.e. Line 223: "[...]and food-insecure 95% CI (5.0% [95% CI: 3.7-6.8])". Please remove the first "95% CI".

4.f. Line 223-224: "[...]than the categories[...]" should be "[...]than in the other categories[...]"

4.g. Line 233: Please consider changing "77% of them on ART" to "77% were on ART"

4.h. Line 233: Please consider changing "had viral suppression" to "were virally suppressed" or "had suppressed viral loads".

4.i. Line 246: You might change "[...]adjusted for ARV detection." to "[...]when adjusted for ARV detection."

4.j. Line 266: I think this is missing a period, and perhaps there should not be a paragraph break here?

4.k. Figure 4: You might remove "based on ARV" since the caption states that these estimates are ARV-adjusted.

7. PLOS authors have the option to publish the peer review history of their article (what does this mean?). If published, this will include your full peer review and any attached files.

Reviewer #1: **Yes: **Joseph Ouma

Reviewer #2: No

---

## [Author Response · Author response to Decision Letter 1]

31 Jan 2022

All the responses are included in rebuttal letter

---

## [Editor Report · Decision Letter 2]

11 Feb 2022

Progress towards controlling the HIV epidemic in urban Ethiopia: Findings from the 2017-2018 Ethiopia population-based HIV impact assessment survey

PONE-D-21-26375R2

Dear Dr. Lulseged,

We’re pleased to inform you that your manuscript has been judged scientifically suitable for publication and will be formally accepted for publication once it meets all outstanding technical requirements.

Kind regards,

David Gerberry, Ph.D.

Academic Editor

PLOS ONE
---

## [Editor Report · Acceptance letter]

16 Feb 2022

PONE-D-21-26375R2 

Progress towards controlling the HIV epidemic in urban Ethiopia: Findings from the 2017-2018 Ethiopia population-based HIV impact assessment survey 

Dear Dr. Lulseged:

I'm pleased to inform you that your manuscript has been deemed suitable for publication in PLOS ONE. Congratulations! Your manuscript is now with our production department. 

Kind regards, 

on behalf of

Dr. David Gerberry 

Academic Editor

PLOS ONE